# How representative are student convenience samples? A study of literacy and numeracy skills in 32 countries

**Heather Wild**[1]*, **Aki-Juhani Kyröläinen**[2], **Victor Kuperman**[1]

**1** Department of Linguistics and Languages, McMaster University, Hamilton, Ontario, Canada, **2** School of Languages and Translation, University of Turku, Turku, Varsinais-Suomi, Finland

* wildh@mcmaster.ca

**Data Availability Statement:** The data underlying the results presented in the study are available from https://www.oecd.org/skills/piaac/data/.

## Abstract

Psychological research, including research into adult reading, is frequently based on convenience samples of undergraduate students. This practice raises concerns about the external validity of many accepted findings. The present study seeks to determine how strong this student sampling bias is in literacy and numeracy research. We use the nationally representative cross-national data from the Programme for the International Assessment of Adult Competencies to quantify skill differences between (i) students and the general population aged 16–65, and (ii) students and age-matched non-students aged 16–25. The median effect size for the comparison (i) of literacy scores across 32 countries was d = .56, and for comparison (ii) d = .55, which exceeds the average effect size in psychological experiments (d = .40). Numeracy comparisons (i) and (ii) showed similarly strong differences. The observed differences indicate that undergraduate students are not representative of the general population nor age-matched non-students.

## Introduction

Over the past two decades growing concerns have been raised about psychological research's overreliance on convenience samples of undergraduate students. Arnett (2008) [1] found that up to 80% of samples in APA-published studies consisted of samples of undergraduate psychology students. A decade later, Rad et al. (2018) [2] reported that although the trend was decreasing, many studies continued to rely on students. Relying heavily on student samples is an extension of the well-known bias of drawing samples from Western Educated Industrialized Rich and Democratic (WEIRD) societies [3]. Not only are student samples frequently drawn from WEIRD countries [1, 2], but they are even WEIRDer within their countries given that students tend to come from higher socio-economic backgrounds, be between age 18–24, and are by nature highly educated. As such, the undergraduate sampling bias compromises one of the core goals of psychological research: external validity.

We are not the first to raise concerns about external validity and the undergraduate sampling bias (see above as well as [4]). Rather, we seek to strengthen the literature by quantifying just how well students represent the general population of their countries. Several previous

**Funding:** This study was supported by the Social Sciences and Humanities Research Council of Canada Partnered Research Training Grant, 895-2016-1008, (Dr. Gary Libben, PI). The first author's contribution was supported by the Social Sciences and Humanities Research Council of Canada's Canada Graduate Scholarship. The second author's contribution was also partially supported by the Social Sciences and Humanities Research Council of Canada Insight Development Grant, 430-2019-00851, (Kyröläinen, PI). The third author's contribution was partially supported by the Canada Research Chair award (Tier 2; Kuperman, PI), and the CFI Leaders Opportunity Fund (Kuperman, PI). The funders had no role in study design, data collection and analysis, decision to publish, or preparation of the manuscript.

**Competing interests:** The authors have declared that no competing interests exist.

studies have found students to be unrepresentative in the field of cognitive psychology. Snowberg and Yariv (2021) [5] found American undergraduates exhibited greater cognitive skill and strategic sophistication than a representative sample of the United States. Similarly, Brañas-Garza et al.'s (2019) [6] cognitive meta-study found students score significantly higher than non-students on the Cognitive Reflection Test (CRT), a measure used to assess decision making processes. Performance on the CRT is also highly correlated with other cognitive measures such as the Wonderlic Personnel Test (WPT), which measures general cognitive ability, and standardized college admissions tests such as the American College Testing (ACT) and Scholastic Aptitude Test (SAT), which measure academic achievement [7]. These findings suggest that relying on undergraduate samples will be equally challenging to the generalizability of educational outcomes such as literacy and numeracy–the focus of this study.

As mentioned above, undergraduate samples do not challenge the generalizability of literacy and numeracy research simply because they are highly educated, but also because they represent a narrow age range. A number of studies indicate age is a significant predictor of cognitive skills including literacy and numeracy. For instance, Kirasic et al. (1996) [8] showed that middle aged and older adults performed worse than young adults on information processing, working memory, and declarative learning tasks, many of which tap into the component skills of literacy and numeracy. Older adults likewise perform worse on direct measures of numeracy skills than younger adults [9–11]. Similarly, Green and Riddell (1998) [12] and Kyröläinen and Kuperman (2021) [13] also report a negative correlation between age and performance on literacy assessments in adults aged 26–65. Therefore, samples of undergraduate students, who tend to be young adults in peak cognitive conditioning, are unlikely to be representative of the cognitive behaviours of the general population.

The current study seeks to quantify just how accurately undergraduate students represent the general population in terms of two complex cognitive skills, namely literacy and numeracy (defined below). There are at least three reasons to single out literacy and numeracy from other cognitive and social phenomena. First, research on these topics is biased towards studying student populations. University students are overrepresented as a source of empirical data in reading research, particularly when it comes to lexical mega-studies and eye-movement corpora. The English Lexicon Project, British Lexicon Project, and Dutch Lexicon Project are large scale collections of lexical decision and naming times for thousands of words in their respective languages and have been used to develop several theories of word processing [14–16]. Similarly, the Ghent Eye-tracking Corpus (GECO) and Multilingual Eye-tracking Corpus (MECO), which recorded eye-movement data while participants read longer texts, have been used to inform theories of reading behaviour and eye-movement control [17, 18]. Each of these valuable and well cited datasets collected their data primarily or exclusively from university students. How well students represent the general population in terms of complex skills such as literacy and numeracy may be an indicator of how representative these samples are in terms of component skills such as reading behaviour, numeric reasoning, working memory, and cognitive control.

The second reason for singling out literacy and numeracy is for their societal importance. In this technological era, these advanced cognitive skills are critical for individual employability, life satisfaction, health, and for the societal and economic prosperity of nations [19, 20]. Finally, literacy and numeracy are skills that students are actively trained on and selected for (e.g. [21]), whereas, on a daily basis, non-students employ these skills to more varied degrees. Over the course of their secondary schooling, individuals typically need to succeed in a series of examinations that precisely target literacy and numeracy in order to be admitted to post-secondary education. Simultaneously, an individual's perception of their literacy and numeracy levels informs their decision on whether to pursue post-secondary education [21]. This

selectivity favors more literate and numerate individuals to become undergraduate students in the first place. In addition, post-secondary education further boosts students' literacy and numeracy by providing intense practice and high stakes for meeting institutional demands on these skills [22, 23]. Against this background, the question is hardly whether university students differ from the broader population of language speakers. Instead, we ask just how different are they?

The present study answers this question by reporting an analysis of literacy and numeracy skills based on comparative data from 24 languages and 32 countries across 5 continents. To our knowledge, this is the first large-scale analysis that quantifies the degree to which undergraduate students represent the general population regarding literacy and numeracy. Given that undergraduate students are the population most frequently sampled in psycholinguistics, we seek to determine how different students are from (i) the general population of adults and (ii) from the age-matched non-student population, in terms of literacy and numeracy skills within and across countries.

We use the Programme for the International Assessment of Adult Competencies (PIAAC) [24] which is an international survey assessing literacy, numeracy and problem-solving skills in the adult population. PIAAC defines literacy as "understanding, evaluating, using and engaging with written texts to participate in society, to achieve one's goals, and to develop one's knowledge and potential" [25]. The PIAAC definition of numeracy is "the ability to access, use, interpret and communicate mathematical information and ideas, in order to engage in and manage the mathematical demands of a range of situations in adult life". The assessment measures reading for a purpose (i.e. to gather knowledge, evaluate the text, form an opinion etc.) [26], see methods for an example. This draws on information processing and working memory skills in addition to basic reading skills such as phonological decoding and vocabulary knowledge. Literacy and numeracy tasks in PIAAC clearly require combining and coordinating multiple cognitive processes and component skills. PIAAC only provides the scores for the most inclusive and complex literacy and numeracy task rather than the individual component skills (except for a small subset of mainly low-literacy participants [27]). Yet group differences in the participants' performance in these complex tasks enables speculation and hypothesis-building with respect to the expected differences in at least some of the required component skills.

One beneficial feature of the PIAAC data is that each participating country was required to produce a probability-based sample (with a minimum size $N = 5000$) representative of the population of adults aged 16 to 65 in the country. Another advantage of the PIAAC data is that the literacy and numeracy scores are psychometrically validated and directly comparable across countries and languages of administration. The result is rich data from 24 languages (including Arabic, Hebrew, Japanese, Kazakh, and Korean) adding valuable insights beyond the over-researched realm of alphabetic Indo-European languages.

## Methods

### Programme for the international assessment of adult competencies

We use the publicly available PIAAC data to estimate effect sizes for comparisons of literacy and numeracy skills between (i) university students and their respective country's adult population (16–65 years old), and (ii) students and non-students in the same age cohort. The more specific comparison (ii) pits undergraduate students against their own age group (16–25 years old) and thus estimates the critical difference while largely subtracting the effect of aging and the cohort effect, which are known to be pivotal in the distribution of cognitive skills in society [13, 28, 29].

We focus on two cognitive skills: literacy and numeracy. Both skills are assessed in PIAAC through tests that simulate the demands of work, social and everyday life on multiple skill facets [30, 31]. For instance, participants may read a list of preschool rules and be asked what is the latest time that children should arrive. In the case of literacy, the test items engage all levels of reading comprehension–including decoding, knowledge of vocabulary, ability to process information at the word, sentence and discourse level, reading fluency and inferential skills–as well as ability to read digital texts (using hyperlinks and navigation). For sample items see http://www.oecd.org/skills/piaac/Literacy%20Sample%20Items.pdf.

The publicly available files with PIAAC data from 35 countries were retrieved from https://www.oecd.org/skills/piaac/data/. We used the files from the first cycle of data collection which took place from 2011–2012 (round 1), 2014–2015 (round 2), and 2017 (round 3). The [redacted] Research Ethics Board deems this use of secondary data exempt from ethics clearance requirements. Three national samples out of the total set of 35 participating countries were removed from the analysis either because they did not contain variables critical for our analyses (Denmark, Russian Federation) or had a sample of fewer than 1000 participants after the trimming described below (Singapore). The following data-processing and trimming steps were applied to the remaining 32 datasets. First, we only considered individuals who were born in the country of test administration and were native speakers of the language in which they took the test. This restriction enabled us to filter out effects of immigration and second language acquisition on the distribution of cognitive skills in a national sample (see [32]). Individual data with missing values for education and occupational status were removed as well.

The resulting national samples and respective weights (see below) were used for estimation of literacy and numeracy skills in different population segments of respective countries. One such segment, labeled Student, included individuals between 16 and 25 years of age who were (a) studying in a formal education setting or working and studying simultaneously, and (b) had completed either upper secondary education, a bachelor's degree or a master's degree at the time of data collection. Another sample, labeled Young, incorporated all individuals in the 16–25 age range who were not part of the Student sample. The final and most inclusive sample, labeled General, consisted of all participants from the trimmed sample of a given country except those in the Student sample. That is, the General sample included the Young sample, but not those in the Student sample. Naturally, many of the participants in the General sample are also former students, which may attenuate the differences between the General and Student samples. Since neither the Young nor the General samples overlapped with the Student sample, we administered pairwise comparisons between independent samples. Sizes of all samples are reported in Table 1 for each country.

## Statistical considerations

Large-scale international assessments such as PIAAC aim to test a broad range of test constructs while minimizing the response burden on the individual. As such, each participant in PIAAC only responded to a subset of test items and a set of plausible values were derived to estimate the individual's overall proficiency, including on the items they did not respond to [33]. The matrix sampling method of PIAAC determines that the sets of items that each participant encounters and responds to are not identical. To enable an accurate estimation of the measurement error, an individual score in each cognitive skill test is represented as 10 plausible estimates of what that person's performance would be. Each plausible value is defined on the test scale from 0 to 500 points. When estimating a participant's performance in, say, a literacy or numeracy task, plausible values are sampled through a bootstrapping procedure to

**Table 1. Descriptive statistics of literacy for the General, Young, and Student national samples, and Cohen's *d* estimates of differences between Student and General samples, and Student and Young samples, presented by country.**

| Ctry | N Total | General | | Student | | Young | | Group comparisons | |
|------|---------|---------|---------|---------|---------|---------|---------|---------|---------|
| | | N | *M(SD)* | N | *M(SD)* | N | *M(SD)* | *d_S_G* | *d_S_Y* |
| AUT | 5130 | 4136 | 273.47 (39.97) | 111 | 301.90 (33.25) | 650 | 279.53 (38.93) | 0.71 | 0.59 |
| BEL | 5463 | 4140 | 277.29 (44.66) | 272 | 301.37 (32.34) | 551 | 281.04 (40.71) | 0.55 | 0.53 |
| CAN | 26683 | 18479 | 278.61 (47.12) | 870 | 295.71 (38.44) | 2653 | 270.88 (45.00) | 0.37 | 0.57 |
| CHL | 5212 | 4680 | 217.33 (52.25) | 316 | 259.64 (41.72) | 696 | 227.43 (47.05) | 0.82 | 0.71 |
| CYP | 5053 | 3687 | 269.95 (39.85) | 207 | 272.76 (33.72) | 395 | 266.72 (37.57) | 0.07 | 0.17 |
| CZE | 6102 | 5398 | 272.96 (40.36) | 453 | 297.03 (35.54) | 986 | 273.01 (38.74) | 0.60 | 0.64 |
| DEU | 5465 | 4395 | 273.77 (45.22) | 228 | 308.00 (33.16) | 696 | 276.87 (42.42) | 0.77 | 0.77 |
| ECU | 5702 | 5166 | 194.96 (50.06) | 301 | 222.69 (42.21) | 1064 | 201.07 (48.00) | 0.56 | 0.46 |
| ESP | 6055 | 4770 | 253.61 (47.26) | 245 | 288.37 (32.06) | 609 | 256.62 (39.39) | 0.75 | 0.85 |
| EST | 7632 | 6124 | 277.36 (43.49) | 379 | 306.77 (34.18) | 919 | 279.61 (40.06) | 0.68 | 0.71 |
| FIN | 5464 | 4899 | 289.59 (46.84) | 234 | 319.76 (34.92) | 611 | 291.90 (38.88) | 0.65 | 0.74 |
| FRA | 6993 | 5620 | 265.61 (45.81) | 357 | 291.21 (38.28) | 672 | 269.10 (42.11) | 0.56 | 0.54 |
| GBR | 8892 | 7361 | 275.39 (47.00) | 379 | 281.94 (39.12) | 708 | 260.60 (44.52) | 0.14 | 0.50 |
| GRC | 4925 | 4198 | 253.57 (46.28) | 273 | 273.74 (41.40) | 323 | 252.68 (45.69) | 0.44 | 0.48 |
| HUN | 6149 | 5665 | 262.73 (45.31) | 207 | 295.28 (31.97) | 648 | 260.86 (41.36) | 0.72 | 0.88 |
| IRL | 5983 | 4543 | 266.58 (46.41) | 181 | 285.49 (35.45) | 418 | 266.31 (40.12) | 0.41 | 0.49 |
| ISR | 5538 | 3706 | 259.23 (54.73) | 217 | 278.85 (40.45) | 897 | 258.92 (48.34) | 0.36 | 0.42 |
| ITA | 4621 | 3909 | 252.19 (43.34) | 153 | 277.16 (37.01) | 322 | 261.65 (39.83) | 0.58 | 0.40 |
| JPN | 5278 | 4906 | 295.83 (39.83) | 231 | 309.44 (31.17) | 528 | 295.87 (35.82) | 0.34 | 0.39 |
| KAZ | 6050 | 3010 | 250.03 (39.04) | 130 | 259.31 (39.57) | 397 | 246.31 (39.91) | 0.24 | 0.33 |
| KOR | 6667 | 6126 | 271.29 (41.00) | 386 | 301.51 (29.80) | 656 | 289.24 (31.41) | 0.75 | 0.40 |
| LTU | 5093 | 4383 | 266.00 (40.74) | 195 | 287.90 (34.20) | 455 | 274.26 (38.34) | 0.54 | 0.37 |
| MEX | 6306 | 5847 | 221.70 (46.40) | 248 | 257.65 (36.65) | 1137 | 228.82 (40.72) | 0.78 | 0.72 |
| NLD | 5170 | 4277 | 288.10 (44.06) | 300 | 315.22 (34.61) | 520 | 287.83 (36.62) | 0.62 | 0.76 |
| NOR | 5128 | 4052 | 283.16 (41.80) | 199 | 300.60 (36.69) | 661 | 273.53 (38.13) | 0.42 | 0.72 |
| NZL | 6177 | 4081 | 283.32 (45.40) | 258 | 294.19 (38.45) | 658 | 274.16 (42.66) | 0.24 | 0.48 |
| PER | 7289 | 6173 | 194.65 (50.68) | 535 | 234.54 (43.63) | 865 | 201.72 (47.09) | 0.80 | 0.72 |
| POL | 9366 | 7263 | 265.17 (47.92) | 1962 | 294.65 (37.71) | 2446 | 274.50 (41.78) | 0.64 | 0.50 |
| SVK | 5723 | 4909 | 274.13 (39.39) | 333 | 291.56 (34.55) | 746 | 270.08 (39.45) | 0.45 | 0.57 |
| SVN | 5331 | 4316 | 258.04 (46.94) | 365 | 287.74 (32.43) | 427 | 264.89 (41.23) | 0.65 | 0.61 |
| SWE | 4469 | 3496 | 288.24 (41.52) | 130 | 307.60 (35.20) | 581 | 287.28 (36.51) | 0.47 | 0.56 |
| USA | 7921 | 5957 | 277.07 (46.36) | 468 | 292.12 (39.56) | 1229 | 270.82 (42.34) | 0.33 | 0.51 |

*M* stands for mean, *SD* for standard deviation, *N* for sample size and *d* for Cohen's *d* (*d*-S-G compares the Student and the General samples; *d*-S-Y the Student and the Young samples). Ctry stands for Country.

produce both a point-wise estimate and an estimate of variability incurred by the non-identical test items that each participant encounters.

Moreover, each participant in the PIAAC survey is associated with a weight, allowing the tested person to stand for a larger segment of the population. The weights are based on census data and determined by the combination of the participant's age, gender, education, place of residence and additional factors (for details see [34]). Specifically, the PIAAC data use Jack-knife Repeated Replication weights that correct for the complex designs of the samples which vary from country to country [34]. Computational procedures have been developed which process the individual plausible values and apply the appropriate weighting to derive estimates

of means and variances that are representative of a given participant sample in the given country (for more detail see [33]). In this analysis, nationally representative estimates of literacy and numeracy have been obtained for the General, Young and Student samples using the package instvy, which is provided in the statistical platform R 3.6.1 [35] and is specifically designed for the PIAAC data [36]. To quantify differences between literacy and numeracy scores between samples, we used the classic Cohen's *d* metric for independent samples, where the difference of means between samples is divided by the pooled standard deviation accounting for nonequal sample sizes [37, 38]. Estimates of Cohen's *d* as an effect size metric are based on estimates of means and standard deviations corrected through weighting to be nationally representative.

## Results

Figs 1 and 2 plot the distribution of literacy and numeracy skills respectively among the Student, Young and General samples of all countries combined. Breakdowns of skill distributions by country can be found in the supplementary materials: S1–S32 Figs for the distributions of literacy skills, and S33–S64 Figs for the distributions of numeracy skills. Notably, both in the

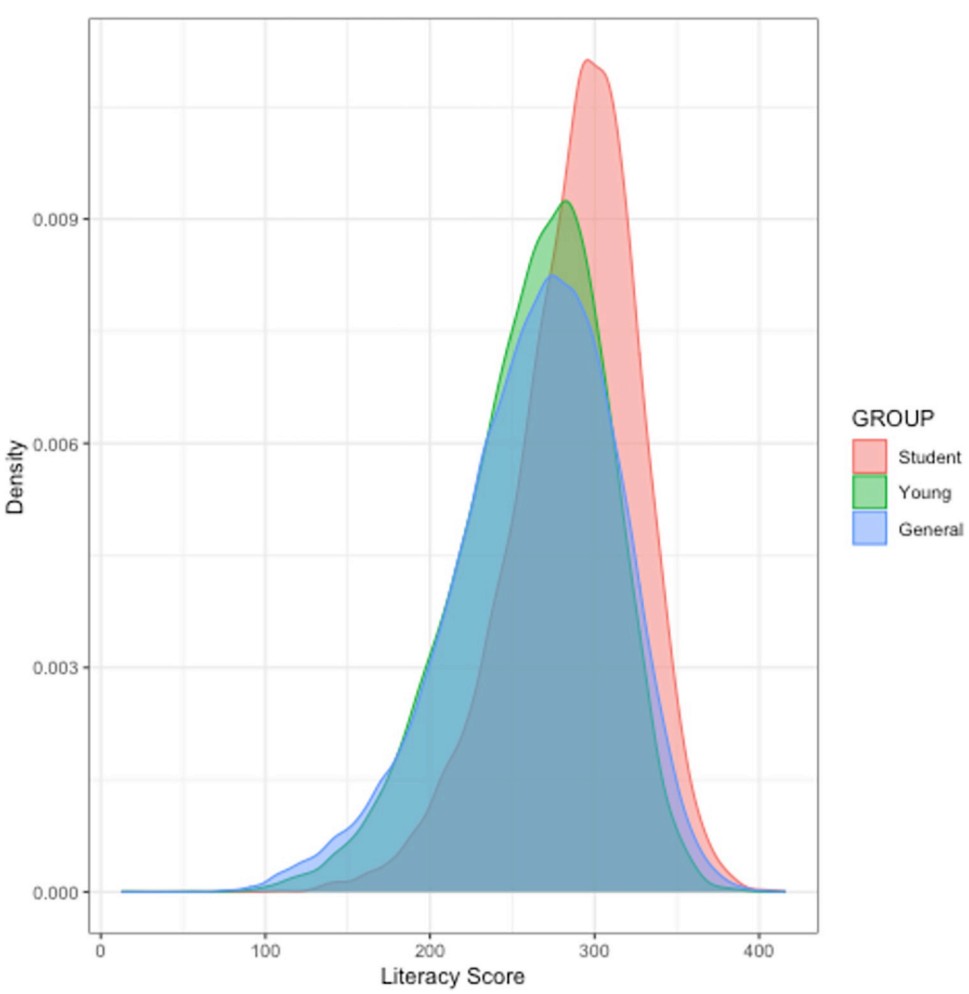

**Fig 1. Distribution of literacy skills among the Student, Young and General samples of all countries.** The red curve represents the Student sample, green represents the Young Sample, and blue represents the General sample.

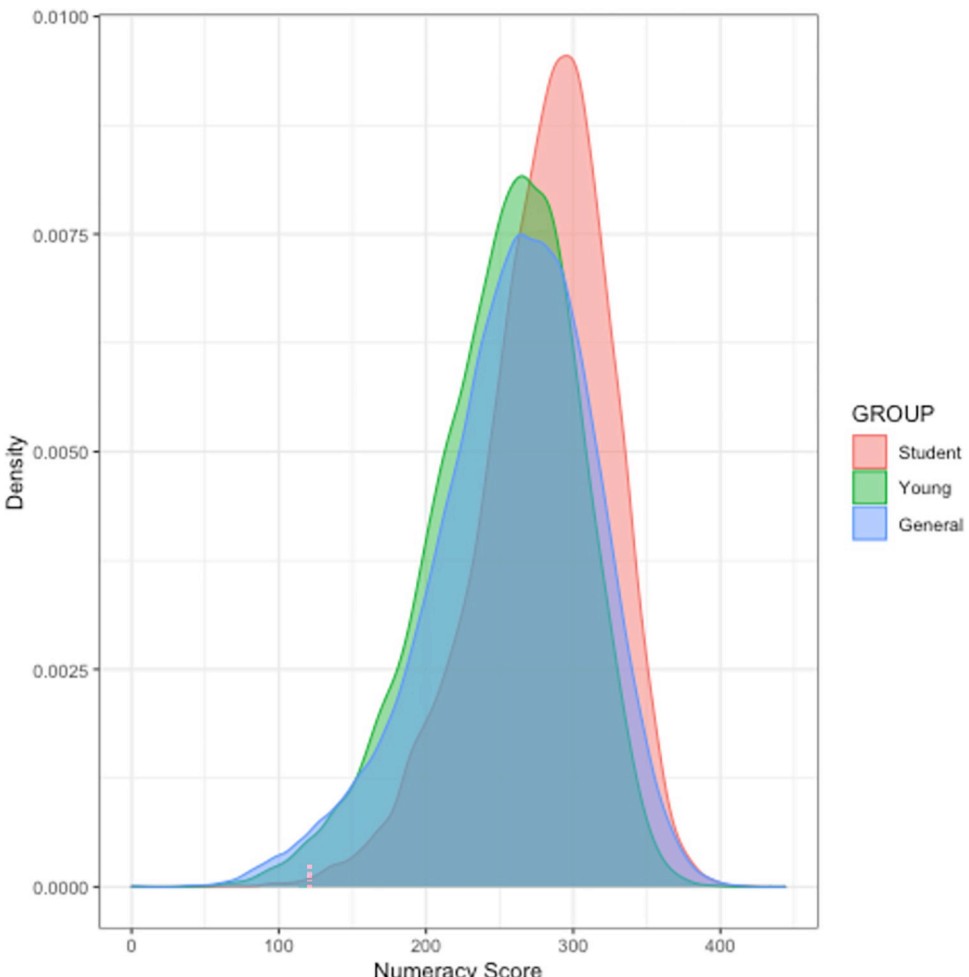

**Fig 2. Distribution of numeracy skills among the Student, Young and General samples of all countries.** The red curve represents the Student sample, green represents the Young Sample, and blue represents the General sample.

aggregated data and in specific countries, the distribution of skills in each sample (General, Student, Young) is symmetrical and the Student sample is shifted to the right relative to the Young and General samples.

Tables 1 and 2 report descriptive statistics and sample sizes for General, Young, and Student samples in each country for literacy and numeracy respectively. Additionally, the Tables report effect sizes (Cohen's *d*) of comparisons between the Student and Young populations, as well as the Student and General populations (see below).

In all countries, the mean literacy and numeracy scores of the Student samples were superior to those found among both young adults and among the general populations. On the PIAAC test scale, the mean difference between Student and General samples was 24 points for literacy and 22 points for numeracy. A comparable advantage of the Student sample over the Young sample was observed: 22 points for literacy and 25 points for numeracy. These differences are massive: that is, they are as large as or larger than the difference between the 25th and 75th percentile of literacy (20 points) and numeracy (23 points) for the General samples of all countries. The variance of scores in the Student sample was not statistically different from variances in either the Young or the General sample, neither in terms of literacy nor numeracy

**Table 2. Descriptive statistics of numeracy for the General, Young, and Student national samples, and Cohen's *d* estimates of differences between Student and General samples, and Student and Young samples, presented by country.**

| Ctry | N Total | General | | Student | | Young | | Group comparisons | |
|---|---|---|---|---|---|---|---|---|---|
| | | N | M(SD) | N | M(SD) | N | M(SD) | d_S_G | d_S_Y |
| AUT | 5130 | 4136 | 280.33 (44.70) | 111 | 304.77 (37.68) | 650 | 281.75 (41.56) | 0.55 | 0.56 |
| BEL | 5463 | 4140 | 282.28 (48.64) | 272 | 300.38 (36.55) | 551 | 278.74 (45.18) | 0.38 | 0.51 |
| CAN | 26683 | 18479 | 269.63 (52.05) | 870 | 288.92 (46.73) | 2653 | 262.42 (51.26) | 0.37 | 0.53 |
| CHL | 5212 | 4680 | 203.24 (58.94) | 316 | 248.59 (43.32) | 696 | 208.92 (49.14) | 0.78 | 0.84 |
| CYP | 5053 | 3687 | 264.56 (46.20) | 207 | 272.14 (39.51) | 395 | 262.06 (43.34) | 0.17 | 0.24 |
| CZE | 6102 | 5398 | 274.97 (43.13) | 453 | 298.47 (36.51) | 986 | 268.82 (41.02) | 0.55 | 0.75 |
| DEU | 5465 | 4395 | 276.41 (50.08) | 228 | 305.91 (38.92) | 696 | 273.23 (45.47) | 0.59 | 0.74 |
| ECU | 5702 | 5166 | 183.50 (53.90) | 301 | 206.19 (45.20) | 1064 | 182.54 (49.10) | 0.42 | 0.49 |
| ESP | 6055 | 4770 | 247.57 (49.80) | 245 | 278.30 (34.24) | 609 | 248.05 (41.72) | 0.63 | 0.76 |
| EST | 7632 | 6124 | 273.79 (44.94) | 379 | 300.48 (36.10) | 919 | 269.90 (41.64) | 0.60 | 0.76 |
| FIN | 5464 | 4899 | 284.63 (48.49) | 234 | 309.67 (39.38) | 611 | 279.25 (42.85) | 0.52 | 0.73 |
| FRA | 6993 | 5620 | 258.89 (52.70) | 357 | 280.89 (42.32) | 672 | 257.15 (48.27) | 0.42 | 0.51 |
| GBR | 8892 | 7361 | 265.80 (52.43) | 379 | 273.34 (42.85) | 708 | 252.28 (47.68) | 0.14 | 0.46 |
| GRC | 4925 | 4198 | 251.18 (48.75) | 273 | 271.04 (41.46) | 323 | 243.90 (46.19) | 0.41 | 0.62 |
| HUN | 6149 | 5665 | 271.25 (52.25) | 207 | 296.61 (40.37) | 648 | 260.70 (47.90) | 0.49 | 0.78 |
| IRL | 5983 | 4543 | 254.61 (52.56) | 181 | 272.99 (44.36) | 418 | 251.94 (46.23) | 0.35 | 0.46 |
| ISR | 5538 | 3706 | 254.68 (63.03) | 217 | 269.98 (49.48) | 897 | 245.94 (54.98) | 0.25 | 0.45 |
| ITA | 4621 | 3909 | 248.49 (49.25) | 153 | 264.10 (42.17) | 322 | 252.58 (45.30) | 0.32 | 0.26 |
| JPN | 5278 | 4906 | 287.77 (44.01) | 231 | 301.36 (38.46) | 528 | 277.10 (41.32) | 0.31 | 0.60 |
| KAZ | 6050 | 3010 | 247.28 (37.24) | 130 | 253.83 (39.23) | 397 | 241.63 (39.14) | 0.18 | 0.31 |
| KOR | 6667 | 6126 | 262.13 (45.27) | 386 | 290.57 (34.40) | 656 | 276.09 (35.58) | 0.64 | 0.41 |
| LTU | 5093 | 4383 | 266.35 (47.24) | 195 | 293.38 (36.73) | 455 | 277.27 (41.54) | 0.58 | 0.40 |
| MEX | 6306 | 5847 | 210.26 (49.69) | 248 | 244.08 (41.11) | 1137 | 214.09 (45.46) | 0.69 | 0.67 |
| NLD | 5170 | 4277 | 285.37 (45.98) | 300 | 307.07 (36.31) | 520 | 278.29 (40.02) | 0.48 | 0.74 |
| NOR | 5128 | 4052 | 284.42 (47.41) | 199 | 300.66 (40.97) | 661 | 269.12 (43.37) | 0.34 | 0.74 |
| NZL | 6177 | 4081 | 272.70 (52.27) | 258 | 284.38 (46.28) | 658 | 261.44 (49.14) | 0.22 | 0.47 |
| PER | 7289 | 6173 | 179.46 (62.74) | 535 | 216.14 (51.00) | 865 | 180.62 (57.77) | 0.59 | 0.64 |
| POL | 9366 | 7263 | 258.22 (50.66) | 1962 | 286.39 (42.36) | 2446 | 259.47 (44.48) | 0.57 | 0.62 |
| SVK | 5723 | 4909 | 276.72 (46.78) | 333 | 293.78 (40.31) | 746 | 272.50 (46.46) | 0.37 | 0.48 |
| SVN | 5331 | 4316 | 260.41 (52.91) | 365 | 289.70 (36.66) | 427 | 263.64 (44.13) | 0.57 | 0.64 |
| SWE | 4469 | 3496 | 288.87 (45.67) | 130 | 302.20 (40.06) | 581 | 283.13 (40.87) | 0.29 | 0.47 |
| USA | 7921 | 5957 | 261.56 (52.77) | 468 | 275.31 (46.70) | 1229 | 251.04 (49.45) | 0.26 | 0.50 |

*M* stands for mean, *SD* for standard deviation, *N* for sample size and *d* for Cohen's *d* (*d*-S-G compares the Student and the General samples; *d*-S-Y the Student and the Young samples). Ctry stands for Country.

(all *F* < 1.3, all *p* > 0.5 in *F* tests). This finding runs counter to the intuition that student samples are more homogenous–due to selectivity of educational institutions and self-selection– than the population at large. However, the finding converges with Hanel and Vione's (2016) [39] report of a similar variability in personality traits and attitudes among students and general populations of 59 countries.

To quantify the differences in a way that is comparable to the relevant psychological literature, we calculated Cohen's *d* metric for independent samples. Cohen's *d* for the comparison of literacy scores between students and the general population ranged from negligible (*d* = 0.07, Cyprus) to strong (*d* = 0.82, Chile), with the median of *d* = 0.56 and *d* = 0.40 and

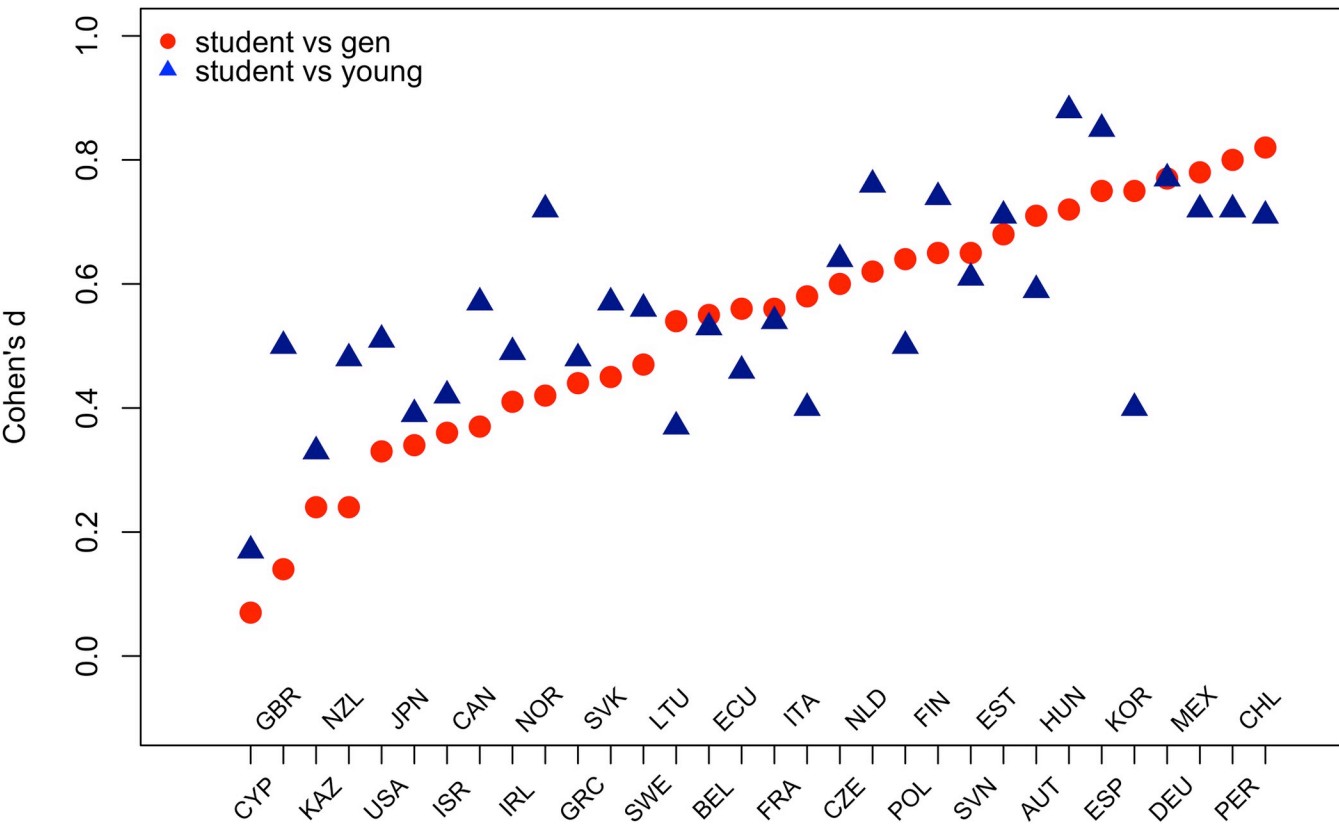

**Fig 3. Effect size comparisons in literacy scores between the Student and General samples and the Student and Young by country.** Red dots represent effect size comparison of Student vs General samples and blue triangles represent effect size comparison for Student vs Young samples.

$d$ = 0.69 as the first and third quartiles. Effect size estimates for differences in literacy between students and young adults were somewhat stronger and less disperse. They ranged from $d$ = 0.17 (Cyprus) to $d$ = 0.88 (Hungary), with the median $d$ = 0.55: the first and third quartiles of this distribution were $d$ = 0.48 and $d$ = 0.71. Fig 3 plots Cohen's $d$ estimates by country in the increasing order of the effect size for the Student vs General group comparison (red dots). Values of Cohen's $d$ for the Student vs Young group comparison are reported in blue triangles.

Comparisons of numeracy scores between students and general population across countries showed Cohen's $d$ values ranging from $d$ = 0.17 (UK) to $d$ = 0.78 (Chile), with the median $d$ = 0.42: the first and third quartiles of this distribution were $d$ = 0.32 and $d$ = 0.57. A comparison of numeracy performance between students and young adults revealed even greater $d$ values. The $d$ values varied between $d$ = 0.24 (Cyprus) and $d$ = 0.84 (Chile), with the median $d$ = 0.55 and $d$ = 0.47 and $d$ = 0.73 as the first and third quartiles. Fig 4 reports Cohen's $d$ estimates for the Student vs General group comparison (red dots) and the Student vs Young group comparison (blue triangles).

The importance of the present findings comes to light when compared against meta-analytical estimates of effect sizes of studies published in the field of psychology. An influential meta-analysis and replication of 100 experimental and correlational papers in psychology [40] places the estimated average effect size of the original studies at $d$ = 0.403 ($SD$ = 0.188) and that of the replications at $d$ = 0.197 ($SD$ = 0.257). Another meta-analysis of 447 psychological papers [41] reports a negative correlation between sample size and effect size. While their estimate of the mean effect size $d$ across all sample sizes is close to 0.4, the largest samples in their data

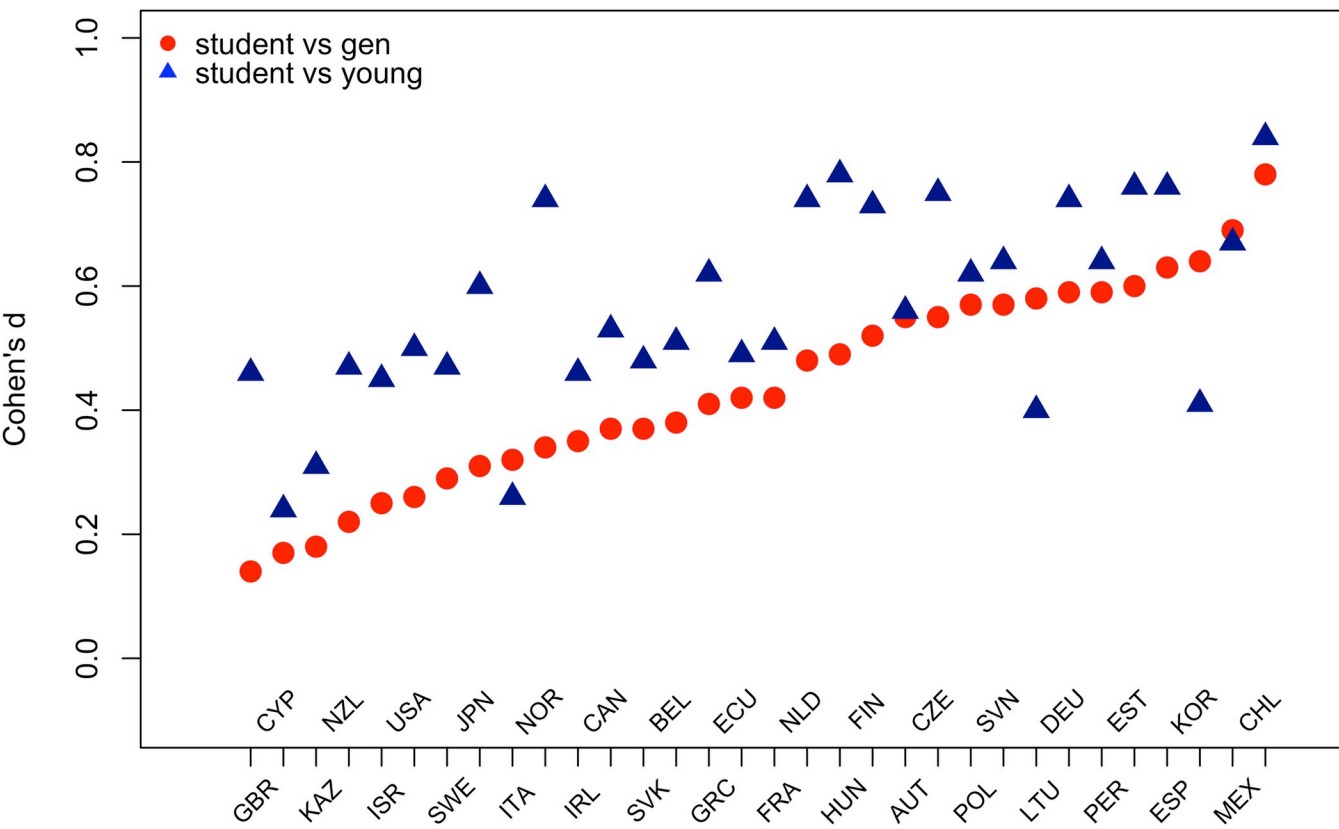

**Fig 4. Effect size comparisons in numeracy scores between the Student and General samples and the Student and Young by country.** Red dots represent effect size comparison of Student vs General samples and blue triangles represent effect size comparison for Student vs Young samples.

($n$ = 500–1000) only yielded a mean effect size $d$ close to 0.25. Since the sample sizes in our data exceed the maximum considered in Kühberger et al. (2014) [41], a predicted effect size for such samples would hover around $d$ = 0.2. Thus, the effect sizes that we observe in our data exceed the expected values by the factor of 2 to 2.5 for both literacy and numeracy when comparing students to both the general and age-matched non-student populations. Moreover, virtually all individual countries in our analyses showed effects stronger than those expected in the published literature in the field of psychology (for variability of effect sizes across types of studies and subdisciplines of psychology see e.g., [42]). In summary, the results show that drawing conclusions about language and math functioning among groups of adult speakers based on evidence from undergraduate students comes with a strong systematic bias in many countries of the world.

## Discussion

The present paper advances the research agenda that examines sampling biases in psychological research, and more specifically literacy and numeracy studies. Convenience samples of undergraduate students are over-represented in the empirical evidence base and play a disproportionately large role in scientific theory-making [1, 2]. Given the common practice of using data from university students to inform theories of linguistic and cognitive processing (as reviewed in the Introduction), reading studies are similarly likely to suffer from a student sampling bias. We quantified just how well undergraduate students represent (i) the general

population (age 16–65) of their country and (ii) age-matched non-students (age 16–25) in terms of literacy and numeracy skills across 32 countries and numerous languages and cultures. Most importantly for the current study, the PIAAC data avoid bias within each selected country. That is, students and all other population segments are represented with the same probability as they naturally occur in that country [43].

In all countries in the dataset, students' mean literacy and numeracy scores were far superior to those of both the non-student young adults and the general populations. While the latter fact may not seem surprising, we find it noteworthy given that many participants in the General population were former post-secondary students and furthered their cognitive skills through additional years of practice. While effect sizes varied across countries, median effect sizes in all comparisons either met or exceeded those typically found in psychological literature ($d > 0.4$) [40–42, 44].

These observations lead to several striking conclusions about the practice of studying language behavior and numeracy using convenience pools of university students. First, it is inaccurate to consider students as a group representative of the population at large. They are as different from the General population (excluding students) as the 25th percentile is different from the 75th percentile in that population. Second, it is even less accurate to treat students as a group representative of non-students of the same age. To put "inaccuracy" into perspective, imagine a high powered pre-registered psychological experiment with a treatment and control group. Imagine further that this group difference shows an effect stronger than those typically observed in experimental psychology ($d > 0.4$). Imagine, finally, that the experimenter interprets the behavior of the treatment group as a valid approximation of the behavior of the control group. In fact, they view the results as support for the null hypothesis and claim the treatment group is representative of the entire population. This scenario is a statistical equivalent of assuming the literacy and numeracy behavior of students represents that of the general or age-matched population of speakers of the same language.

One clear theoretical impact of this mismatch between students' reading skills and those of other populations is that it raises questions about generalizability and external validity of empirical research based on the findings from literacy- and numeracy-related behaviours of undergraduate students. To be clear, sampling from student populations is not in itself a problem, so long as the findings are interpreted within the student population. Yet such disclaimers are rarely found in psychological literature (including in our own work). Consequently, readers can make the logical assumption that findings based on undergraduate student groups generalize over the entire adult population. However, as the findings above indicate, students are rarely representative of the adult population when it comes to literacy and numeracy. Therefore, we hope to have demonstrated that caution is needed when studying phenomena that rely on highly trained cognitive skills such as literacy and numeracy.

## Limitations and future directions

The estimates in this study are calculated on the basis of a single, though complex and comprehensive, task. As such, we can only say with certainty that students do not represent other populations in terms of the PIAAC measures of literacy and numeracy. It is up to future research to quantify how representative undergraduate students are on other measures of literacy and numeracy. For instance, we speculate that students will not be representative of other population groups in terms of their fluency in literacy and numeracy-related tasks. Specifically, we predict students to be faster than other populations both because they showed higher accuracy in the tasks reported here, and because of multiple reports of higher being associated with higher speed of task completion: for early reports and recent reviews in reading see [45, 46].

Literacy and numeracy, particularly as assessed in PIAAC, require the coordination of multiple cognitive processes and mastery of multiple component skills. We predict that students will also be unrepresentative when it comes to the component skills of reading and numeracy such as working memory, numeric reasoning, and word processing. Since PIAAC does not test these component skills directly, the current study cannot indicate whether these group differences indeed exist or whether the effect sizes will be reduced or amplified on other tasks. Future investigations should continue to quantify differences between student and other populations both on comprehensive literacy and numeracy assessments, as well as tasks targeting their component skills. This paper provides a qualitative indication that such differences are likely to be found.

The main question explored in this study–how different are students' cognitive skills from those of other population groups–is coupled with at least two other questions that are out of the scope of the present paper: (a) what contributes to these differences, and (b) how do these differences influence the inquiry into psychological traits and processes in other domains. Question (a) has been extensively covered in studies of literacy and numeracy development as well as research on post-secondary education (for select reviews see [21, 47–49]). We note however that the by-country breakdown of the differences between samples (reported in Tables 1 and 2) can further boost this research as these differences are likely to be co-determined by demographic and socio-economic characteristics of those countries and their investment in both the spread and quality of (post-secondary) education. The present data do not shed light on question (b), therefore we relegate further exploration of (a) and (b) to future research. We also note that the present study highlighted group differences in advanced behaviors tested in PIAAC data. These behaviors demand a proficient and coordinated use of multiple component skills, including word recognition, reading fluency and reading comprehension. How the over-reliance on sampling university students affects the accuracy of verbal and computational models of such component skills (partly discussed in the Introduction) is an important question for further examination.

## Conclusion

To be sure, few researchers of literacy or numeracy are likely to endorse a premise that students accurately represent the literacy or numeracy skills of the general population. Yet it is important to realize that this premise is implicit in the common practice of reporting experimental findings or computational models based on university students without a disclaimer about their limited generalizability. We do not wish to imply that the field of language or numeracy research is ignorant of the problem. To give only a few examples to the contrary, there are ongoing efforts to study literacy in older adults [8, 12, 50], communities of low socio-economic status [51], as well as readers with lower literacy or lower academic attainment populations [20, 52–58]. Additionally, an increasing number of comparative literacy and numeracy studies draw community or representative samples for their hypothesis testing (see among many others [13, 29, 59–62]. Finally, as undergraduate sampling relates to the WEIRD bias, we also highlight the growing body of cross-linguistically comparable samples in literacy research (among others, [18, 63–66].

Still, collecting normative population-wide data is an expensive, time-consuming process, and funding agencies can be more reluctant to provide support for such endeavors than for research of groups defined by their clinical, demographic, or social status. Change in the culture of research must be complemented by change in scientific policy-making. We echo the recommendations of Henrich et al. (2010) [3] and Rad et al. (2018) [2] for researchers to explicitly address questions of generalizability in their samples, make data freely available to

aid comparative research efforts, collect data broadly within their countries, and build partnerships with community members and researchers, particularly in non-WEIRD countries. Moreover, we urge funding agencies and policy-makers to recognise the importance of minimising the student sampling bias in language research and value projects with representative and non-WEIRD samples accordingly. The movement towards more inclusive data coverage and external support for such coverage is necessary to maintain high standards of psychological research.

## Supporting information

**S1 Fig. Distribution of literacy skills among the Student, Young and General samples of Austria.** The red curve represents the Student sample, green represents the Young Sample, and blue represents the General sample.
(TIF)

**S2 Fig. Distribution of literacy skills among the Student, Young and General samples of Belgium.** The red curve represents the Student sample, green represents the Young Sample, and blue represents the General sample.
(TIF)

**S3 Fig. Distribution of literacy skills among the Student, Young and General samples of Canada.** The red curve represents the Student sample, green represents the Young Sample, and blue represents the General sample.
(TIF)

**S4 Fig. Distribution of literacy skills among the Student, Young and General samples of Chile.** The red curve represents the Student sample, green represents the Young Sample, and blue represents the General sample.
(TIF)

**S5 Fig. Distribution of literacy skills among the Student, Young and General samples of Cyprus.** The red curve represents the Student sample, green represents the Young Sample, and blue represents the General sample.
(TIF)

**S6 Fig. Distribution of literacy skills among the Student, Young and General samples of Czech Republic.** The red curve represents the Student sample, green represents the Young Sample, and blue represents the General sample.
(TIF)

**S7 Fig. Distribution of literacy skills among the Student, Young and General samples of Germany.** The red curve represents the Student sample, green represents the Young Sample, and blue represents the General sample.
(TIF)

**S8 Fig. Distribution of literacy skills among the Student, Young and General samples of Ecuador.** The red curve represents the Student sample, green represents the Young Sample, and blue represents the General sample.
(TIF)

**S9 Fig. Distribution of literacy skills among the Student, Young and General samples of Spain.** The red curve represents the Student sample, green represents the Young Sample, and blue represents the General sample.
(TIF)

**S10 Fig. Distribution of literacy skills among the Student, Young and General samples of Estonia.** The red curve represents the Student sample, green represents the Young Sample, and blue represents the General sample.
(TIF)

**S11 Fig. Distribution of literacy skills among the Student, Young and General samples of Finland.** The red curve represents the Student sample, green represents the Young Sample, and blue represents the General sample.
(TIF)

**S12 Fig. Distribution of literacy skills among the Student, Young and General samples of France.** The red curve represents the Student sample, green represents the Young Sample, and blue represents the General sample.
(TIF)

**S13 Fig. Distribution of literacy skills among the Student, Young and General samples of Great Britain.** The red curve represents the Student sample, green represents the Young Sample, and blue represents the General sample.
(TIF)

**S14 Fig. Distribution of literacy skills among the Student, Young and General samples of Greece.** The red curve represents the Student sample, green represents the Young Sample, and blue represents the General sample.
(TIF)

**S15 Fig. Distribution of literacy skills among the Student, Young and General samples of Hungary.** The red curve represents the Student sample, green represents the Young Sample, and blue represents the General sample.
(TIF)

**S16 Fig. Distribution of literacy skills among the Student, Young and General samples of Ireland.** The red curve represents the Student sample, green represents the Young Sample, and blue represents the General sample.
(TIF)

**S17 Fig. Distribution of literacy skills among the Student, Young and General samples of Israel.** The red curve represents the Student sample, green represents the Young Sample, and blue represents the General sample.
(TIF)

**S18 Fig. Distribution of literacy skills among the Student, Young and General samples of Italy.** The red curve represents the Student sample, green represents the Young Sample, and blue represents the General sample.
(TIF)

**S19 Fig. Distribution of literacy skills among the Student, Young and General samples of Japan.** The red curve represents the Student sample, green represents the Young Sample, and blue represents the General sample.
(TIF)

**S20 Fig. Distribution of literacy skills among the Student, Young and General samples of Kazakhstan.** The red curve represents the Student sample, green represents the Young

Sample, and blue represents the General sample.
(TIF)

**S21 Fig. Distribution of literacy skills among the Student, Young and General samples of Korea.** The red curve represents the Student sample, green represents the Young Sample, and blue represents the General sample.
(TIF)

**S22 Fig. Distribution of literacy skills among the Student, Young and General samples of Lithuania.** The red curve represents the Student sample, green represents the Young Sample, and blue represents the General sample.
(TIF)

**S23 Fig. Distribution of literacy skills among the Student, Young and General samples of Mexico.** The red curve represents the Student sample, green represents the Young Sample, and blue represents the General sample.
(TIF)

**S24 Fig. Distribution of literacy skills among the Student, Young and General samples of Netherlands.** The red curve represents the Student sample, green represents the Young Sample, and blue represents the General sample.
(TIF)

**S25 Fig. Distribution of literacy skills among the Student, Young and General samples of New Zealand.** The red curve represents the Student sample, green represents the Young Sample, and blue represents the General sample.
(TIF)

**S26 Fig. Distribution of literacy skills among the Student, Young and General samples of Norway.** The red curve represents the Student sample, green represents the Young Sample, and blue represents the General sample.
(TIF)

**S27 Fig. Distribution of literacy skills among the Student, Young and General samples of Peru.** The red curve represents the Student sample, green represents the Young Sample, and blue represents the General sample.
(TIF)

**S28 Fig. Distribution of literacy skills among the Student, Young and General samples of Poland.** The red curve represents the Student sample, green represents the Young Sample, and blue represents the General sample.
(TIF)

**S29 Fig. Distribution of literacy skills among the Student, Young and General samples of Slovakia.** The red curve represents the Student sample, green represents the Young Sample, and blue represents the General sample.
(TIF)

**S30 Fig. Distribution of literacy skills among the Student, Young and General samples of Slovenia.** The red curve represents the Student sample, green represents the Young Sample, and blue represents the General sample.
(TIF)

**S31 Fig. Distribution of literacy skills among the Student, Young and General samples of Sweden.** The red curve represents the Student sample, green represents the Young Sample, and blue represents the General sample.
(TIF)

**S32 Fig. Distribution of literacy skills among the Student, Young and General samples of United States of America.** The red curve represents the Student sample, green represents the Young Sample, and blue represents the General sample.
(TIF)

**S33 Fig. Distribution of numeracy skills among the Student, Young and General samples of Austria.** The red curve represents the Student sample, green represents the Young Sample, and blue represents the General sample.
(TIF)

**S34 Fig. Distribution of numeracy skills among the Student, Young and General samples of Belgium.** The red curve represents the Student sample, green represents the Young Sample, and blue represents the General sample.
(TIF)

**S35 Fig. Distribution of numeracy skills among the Student, Young and General samples of Canada.** The red curve represents the Student sample, green represents the Young Sample, and blue represents the General sample.
(TIF)

**S36 Fig. Distribution of numeracy skills among the Student, Young and General samples of Chile.** The red curve represents the Student sample, green represents the Young Sample, and blue represents the General sample.
(TIF)

**S37 Fig. Distribution of numeracy skills among the Student, Young and General samples of Cyprus.** The red curve represents the Student sample, green represents the Young Sample, and blue represents the General sample.
(TIF)

**S38 Fig. Distribution of numeracy skills among the Student, Young and General samples of Czech Republic.** The red curve represents the Student sample, green represents the Young Sample, and blue represents the General sample.
(TIF)

**S39 Fig. Distribution of numeracy skills among the Student, Young and General samples of Germany.** The red curve represents the Student sample, green represents the Young Sample, and blue represents the General sample.
(TIF)

**S40 Fig. Distribution of numeracy skills among the Student, Young and General samples of Ecuador.** The red curve represents the Student sample, green represents the Young Sample, and blue represents the General sample.
(TIF)

**S41 Fig. Distribution of numeracy skills among the Student, Young and General samples of Spain.** The red curve represents the Student sample, green represents the Young Sample, and blue represents the General sample.
(TIF)

**S42 Fig. Distribution of numeracy skills among the Student, Young and General samples of Estonia.** The red curve represents the Student sample, green represents the Young Sample, and blue represents the General sample.
(TIF)

**S43 Fig. Distribution of numeracy skills among the Student, Young and General samples of Finland.** The red curve represents the Student sample, green represents the Young Sample, and blue represents the General sample.
(TIF)

**S44 Fig. Distribution of numeracy skills among the Student, Young and General samples of France.** The red curve represents the Student sample, green represents the Young Sample, and blue represents the General sample.
(TIF)

**S45 Fig. Distribution of numeracy skills among the Student, Young and General samples of Great Britain.** The red curve represents the Student sample, green represents the Young Sample, and blue represents the General sample.
(TIF)

**S46 Fig. Distribution of numeracy skills among the Student, Young and General samples of Greece.** The red curve represents the Student sample, green represents the Young Sample, and blue represents the General sample.
(TIF)

**S47 Fig. Distribution of numeracy skills among the Student, Young and General samples of Hungary.** The red curve represents the Student sample, green represents the Young Sample, and blue represents the General sample.
(TIF)

**S48 Fig. Distribution of numeracy skills among the Student, Young and General samples of Ireland.** The red curve represents the Student sample, green represents the Young Sample, and blue represents the General sample.
(TIF)

**S49 Fig. Distribution of numeracy skills among the Student, Young and General samples of Israel.** The red curve represents the Student sample, green represents the Young Sample, and blue represents the General sample.
(TIF)

**S50 Fig. Distribution of numeracy skills among the Student, Young and General samples of Italy.** The red curve represents the Student sample, green represents the Young Sample, and blue represents the General sample.
(TIF)

**S51 Fig. Distribution of numeracy skills among the Student, Young and General samples of Japan.** The red curve represents the Student sample, green represents the Young Sample, and blue represents the General sample.
(TIF)

**S52 Fig. Distribution of numeracy skills among the Student, Young and General samples of Kazakhstan.** The red curve represents the Student sample, green represents the Young

Sample, and blue represents the General sample.
(TIF)

**S53 Fig. Distribution of numeracy skills among the Student, Young and General samples of Korea.** The red curve represents the Student sample, green represents the Young Sample, and blue represents the General sample.
(TIF)

**S54 Fig. Distribution of numeracy skills among the Student, Young and General samples of Lithuania.** The red curve represents the Student sample, green represents the Young Sample, and blue represents the General sample.
(TIF)

**S55 Fig. Distribution of numeracy skills among the Student, Young and General samples of Mexico.** The red curve represents the Student sample, green represents the Young Sample, and blue represents the General sample.
(TIF)

**S56 Fig. Distribution of numeracy skills among the Student, Young and General samples of Netherlands.** The red curve represents the Student sample, green represents the Young Sample, and blue represents the General sample.
(TIF)

**S57 Fig. Distribution of numeracy skills among the Student, Young and General samples of New Zealand.** The red curve represents the Student sample, green represents the Young Sample, and blue represents the General sample.
(TIF)

**S58 Fig. Distribution of numeracy skills among the Student, Young and General samples of Norway.** The red curve represents the Student sample, green represents the Young Sample, and blue represents the General sample.
(TIF)

**S59 Fig. Distribution of numeracy skills among the Student, Young and General samples of Peru.** The red curve represents the Student sample, green represents the Young Sample, and blue represents the General sample.
(TIF)

**S60 Fig. Distribution of numeracy skills among the Student, Young and General samples of Poland.** The red curve represents the Student sample, green represents the Young Sample, and blue represents the General sample.
(TIF)

**S61 Fig. Distribution of numeracy skills among the Student, Young and General samples of Slovakia.** The red curve represents the Student sample, green represents the Young Sample, and blue represents the General sample.
(TIF)

**S62 Fig. Distribution of numeracy skills among the Student, Young and General samples of Slovenia.** The red curve represents the Student sample, green represents the Young Sample, and blue represents the General sample.
(TIF)

**S63 Fig. Distribution of numeracy skills among the Student, Young and General samples of Sweden.** The red curve represents the Student sample, green represents the Young Sample, and blue represents the General sample.
(TIF)

**S64 Fig. Distribution of numeracy skills among the Student, Young and General samples of United States of America.** The red curve represents the Student sample, green represents the Young Sample, and blue represents the General sample.
(TIF)

## Acknowledgments

We analyzed publicly available data that are not under our direct control; the data can be accessed at https://www.oecd.org/skills/piaac/data/. Excerpts from this paper and a published one-page abstract were presented at the Words in the World Virtual Conference in 2020. We thank the reviewers for their insightful comments and feedback.

## Author Contributions

**Conceptualization:** Heather Wild.

**Formal analysis:** Heather Wild, Victor Kuperman.

**Methodology:** Heather Wild, Aki-Juhani Kyröläinen, Victor Kuperman.

**Software:** Aki-Juhani Kyröläinen.

**Writing – original draft:** Heather Wild, Victor Kuperman.

**Writing – review & editing:** Heather Wild, Aki-Juhani Kyröläinen, Victor Kuperman.

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
