## [Decision Letter · Decision Letter 0]

21 Apr 2022

PONE-D-22-04101Just how misleading are student convenience samples? A comparative study of literacy and numeracy in 32 countriesPLOS ONE

Dear Dr. Wild,

Thank you for submitting your manuscript to PLOS ONE. After careful consideration, we feel that it has merit but does not fully meet PLOS ONE’s publication criteria as it currently stands. Therefore, we invite you to submit a revised version of the manuscript that addresses the points raised during the review process.

Two expert Reviewers have now carefully scrutinised your submission (Reviewer 1 chose to remain anonymous, Reviewer 2 is Dr Julia Carroll). In addition, I have read the manuscript with great interest as it deals with a topic close to my heart, and have voiced similar concerns for some time. Both Reviewers and I believe that your manuscript deals with an important and timely topic, and I am of the opinion that a suitably revised version of it would certainly be of interest to the scientific community. However, both Reviewers have highlighted a number of points which need to be taken into account in your revision, and I would like you to address each one of them as I believe they are all (very) valid concerns/suggestions.

In line with Reviewer's 1 comment on simulations, I too found the diversion into the modelling part somewhat puzzling as EZ Reader deals with reading time data, which is not tested in the PIAAC. Moreover, while the original parameter settings might have been based on a restricted sample, there is a large literature that show that the model does remarkably well in simulating actual (undergraduate) reading data. Personally, I would suggest to de-emphasise the somewhat hazy link with the modelling literature and provide more detail on important differences that have been found when more diverse populations have been included (see references on lines 298-303). 

Thank you for submitting this interesting study and I am looking forward to reading a revised version of the manuscript.

Steven Frisson

We look forward to receiving your revised manuscript.

Kind regards,

Steven Frisson

Academic Editor

PLOS ONE

Journal Requirements:

Reviewers' comments:

Reviewer's Responses to Questions

**Comments to the Author**

1. Is the manuscript technically sound, and do the data support the conclusions?

Reviewer #1: Partly

Reviewer #2: Yes

2. Has the statistical analysis been performed appropriately and rigorously? 

Reviewer #1: N/A

Reviewer #2: Yes

3. Have the authors made all data underlying the findings in their manuscript fully available?

Reviewer #1: Yes

Reviewer #2: Yes

4. Is the manuscript presented in an intelligible fashion and written in standard English?

Reviewer #1: Yes

Reviewer #2: Yes

5. Review Comments to the Author

Reviewer #1: Using data from PIAAC, the authors tested whether undergraduate students are representative of the general population and/or of age-matched controls in terms of literacy and numeracy. The authors report large differences between undergraduate students and both the general population and age-matched controls, indicating that findings from studies that use convenience samples of undergraduate students cannot necessarily be expected to generalize.

I like this paper for its briefness and clarity, as well as for its potentially very important conclusion. Nevertheless, I believe that some points need to be addressed by the authors before publication:

* The title is too strong given the conclusions. The title presupposes that undergraduate samples *are* misleading - something that doesn't necessarily follow from the results, given that, for instance, it is never demonstrated that numeracy actually influences reading behavior (see below). It depends entirely on what is being studied.

* There is something of a disconnect between the literature review on page 2 and the discussion of psycholinguistic computational modeling on page 3. It is left to the reader to think about how differences in personality measures or analytic thinking would cast doubt on the generalizability of parameter estimates in the E-Z Reader model, for instance. It may help to briefly introduce some of the relevant parameters and speculate about connections here.

* More generally, I wasn't quite sure where the strong initial focus on computational modeling was coming from. The main point - that conclusions drawn from student data may not generalize to the overall population - applies to any study that uses undergraduate students as "controls", even to studies where, say, monolingual undergraduate students are compared to bilingual undergraduate students. It is, for instance, conceivable that the student status of the sampled individuals would nullify any advantages or disadvantages that bilinguals may have compared to monolinguals.

* This ties into my next point of criticism: The authors largely avoid speculating about the possible effects that differences in numeracy and literacy as measured by PIAAC may have on, say, online reading behavior. I was not able to find out if PIAAC measured reading times to assess reading fluency, for instance. I understand that it's not the authors' main interest to develop a theory of what connections may exist, but I think readers would welcome some speculation in this regard. This is perhaps even more important for numeracy, where I can't think of any obvious connections from the top of my head.

* page 2: CRT, ACT, SAT and WPT should be briefly explained.

* page 2, line 59, "The second reason for singling out ..." - The scientific relevance of this point is not clear to me.

* page 4, line 67: "Individuals typically need to succeed ..." - This sounds like it's referring to university admission tests, which are not used everywhere in the world. I assume that the authors are also referring more generally to any tests taken during secondary education that influence people's chances of getting admitted to university, but this should be clarified.

* page 5, line 115: I looked at the PIAAC sample items, which I had never seen before. What struck me is that the sample items are very specific in that a clear question with a relatively simple, unambiguous answer is asked before participants even start reading. The second item is also not a text but a table. In both cases, 95% of the contents can be ignored by the reader if they only want to answer the question. What I'm asking myself is how representative these sample items are of PIAAC as a whole, and how similar the PIAAC items are to items used in, say, a typical psycholinguistic experiment. This is relevant because the authors' contention is that people who perform differently from the general population in PIAAC can't be reasonably expected to perform similarly to the general population in a psycholinguistic experiment - but if the two tasks mostly tap into unrelated skills, this conclusion would be less justified.

* page 6: The "Statistical considerations" section needs some clarification. The section starts with the introduction of "10 plausible values", which the reader doesn't know what to do with at this point. There is some explanation later on, but it doesn't quite become clear to the uninitiated reader what is happening and why the raw data can't be used.

* page 8: Instead of Tables 1 and 2, I would have liked to see a histogram of the distribution of literacy and numeracy scores in each sample. For instance, are the scores normally distributed? This is crucial for interpreting mean and SD values. Furthermore, I don't quite see what showing the scores for each country contributes to the main point.

* page 10: I'm not sure that the comparison of effect sizes between PIAAC score differences and psychology studies is warranted or useful. Even though Cohen's d is supposed to abstract away from differences wrt *what* was measured on which scale, I think it's still apples and oranges to compare differences in traits between groups of people with differences in dependent variables caused by some experimental manipulation. I think this part could be dropped without loss of clarity or relevance. By contrast, what I would like to see are the estimates of the score differences between groups and their associated confidence intervals. In order to interpret these, it would be helpful to know what the distribution of scores is (see previous point).

* page 11: "We identified the magnitude of this bias ... " - This needs clarification. The authors did not identify the magnitude of the undergraduate sampling bias, which to my mind would indicate how much more likely undergraduates are to be sampled compared to other people. I think what the authors mean is that they quantified the sampling bias regarding literacy and numeracy skills that *results* from the undergraduate sampling bias.

* References: Many references have incorrect capitalization, incomplete page numbers, and other formatting issues.

Reviewer #2: This paper is clear and straightforward, raises an important point, and uses good quality data and analyses. I have only minor queries. Frist, the description of the three different groups on page 6 (line 141) makes it sound like participants could not be part of the 'young' group and the 'general' group, but in actual fact I think from the description that these are overlapping samples. This should be clarified. Second, I felt that is worth emphasising that these estimates have been calculated on the basis of single, relatively complex tasks. It is therefore not possible to know whether similar effect sizes exist for different types of literacy measure or numeracy measure. For example, it may be that group differences are reduced for simple tasks such as word reading, but the authors are not able to confirm this.

6. PLOS authors have the option to publish the peer review history of their article (what does this mean?). If published, this will include your full peer review and any attached files.

Reviewer #1: No

Reviewer #2: **Yes: **Dr Julia Carroll

---

## [Author Response · Author response to Decision Letter 0]

30 May 2022

We thank the editor and reviewers for their insightful comments and appreciate their feedback. We feel this revised manuscript is a clearer and more streamlined discussion of how well undergraduate students represent the general population in terms of literacy and numeracy. Below we have addressed each comment individually.

Editor:

In line with Reviewer's 1 comment on simulations, I too found the diversion into the modelling

part somewhat puzzling as EZ Reader deals with reading time data, which is not tested in the

PIAAC. Moreover, while the original parameter settings might have been based on a restricted

sample, there is a large literature that show that the model does remarkably well in simulating

actual (undergraduate) reading data. Personally, I would suggest to de-emphasise the

somewhat hazy link with the modelling literature and provide more detail on important

differences that have been found when more diverse populations have been included (see

references on lines 298-303).

We thank the editor for this point. We have removed the modelling literature from the introduction and de-emphasised it in the discussion. We agree that this deletion makes for a more streamlined argument. This section of the introduction (lines 83 – 99) now focuses on how several large datasets such as the British Lexicon Project and Ghent Eye-tracking Corpus, which inform theories of word processing and reading behaviour, collect data primarily or exclusively from undergraduate student samples. We make the link that if students are unrepresentative when it comes to complex skills like numeracy and literacy, they are also likely to be unrepresentative in terms of component skills such as reading behaviour, numeric reasoning, working memory, and cognitive control. In addition, more examples of differences between student and the general population have been included on lines 49 – 82.

Reviewer 1: 

* The title is too strong given the conclusions. The title presupposes that undergraduate

samples *are* misleading - something that doesn't necessarily follow from the results, given

that, for instance, it is never demonstrated that numeracy actually influences reading behavior

(see below). It depends entirely on what is being studied.

We revised the title to “How representative are student convenience samples? A study of literacy and numeracy skills in 32 countries”. The title no longer presupposes student samples are misleading. To clarify, the study is not correlating literacy and numeracy nor stating that numeracy influences reading behaviour. Rather we ask how well students’ literacy represents other populations’ literacy levels and, separately, how well students’ numeracy represents other populations’ numeracy levels.

* There is something of a disconnect between the literature review on page 2 and the

discussion of psycholinguistic computational modeling on page 3. It is left to the reader to think

about how differences in personality measures or analytic thinking would cast doubt on the

generalizability of parameter estimates in the E-Z Reader model, for instance. It may help to

briefly introduce some of the relevant parameters and speculate about connections here.

We thank the reviewer for this point. As noted above in response to the Editor’s comments, we have largely re-worked these sections of the introduction (lines 37 – 99). Discussion of personality measures has been removed and the literature on page 2 now focuses mainly on cognitive differences between undergraduate students and the general population. Lines 37 – 82 make a more explicit link between general cognitive skills and literacy/numeracy in that the tasks used to measure many cognitive skills target the component skills of literacy/numeracy. This is echoed again on lines 95 – 99. 

* More generally, I wasn't quite sure where the strong initial focus on computational modeling

was coming from. The main point - that conclusions drawn from student data may not

generalize to the overall population - applies to any study that uses undergraduate students as

"controls", even to studies where, say, monolingual undergraduate students are compared to

bilingual undergraduate students. It is, for instance, conceivable that the student status of the

sampled individuals would nullify any advantages or disadvantages that bilinguals may have

compared to monolinguals.

The discussion on computational models has been removed from the present manuscript. Indeed, we agree that this removal makes for a more streamlined argument. Lines 87 – 99 now discuss how several large datasets such as the British Lexicon Project and Ghent Eye-tracking corpus, which inform theories of word processing and reading behaviour, collect data primarily or exclusively from undergraduate student samples. We make the link that if students are unrepresentative when it comes to complex skills like numeracy and literacy, they are also likely to be unrepresentative in terms of component skills such as reading behaviour, numeric reasoning, working memory, and cognitive control. 

We also agree with the reviewer’s analogy with monolingual vs bilingual students and would like to develop it further. If it were the case that researchers would only use convenience samples of monolingual students and claim their behavior to be representative of all students, they would introduce a systematic bias in their estimates. This is analogous to using all students (monolingual or bilingual) and presenting their performance as representative of that of the general population (monolingual or bilingual). It is also worth a reminder that in the present study we only considered native speakers of the respective languages of testing.

* This ties into my next point of criticism: The authors largely avoid speculating about the

possible effects that differences in numeracy and literacy as measured by PIAAC may have on,

say, online reading behavior. I was not able to find out if PIAAC measured reading times to

assess reading fluency, for instance. I understand that it's not the authors' main interest to

develop a theory of what connections may exist, but I think readers would welcome some

speculation in this regard. This is perhaps even more important for numeracy, where I can't

think of any obvious connections from the top of my head.

While PIAAC does collect reading time data, this information is not available in the public data which we used in this study. We added a speculation in the discussion (lines 460-472), based on existing evidence from the reading literature, that highly proficient readers (like students) are faster than less proficient ones in reading for comprehension. This contrast is likely to propagate to the PIAAC tests of literacy, such that reading times estimates based on students will be unrepresentative (faster) than those of the general population or non-students of the matched age. 

* page 2: CRT, ACT, SAT and WPT should be briefly explained.

Explanations added on lines 42 – 46.

* page 2, line 59, "The second reason for singling out ..." - The scientific relevance of this point

is not clear to me.

Points 1 and 3 make a scientific argument for our interest in literacy and numeracy skills. This point is intended to highlight the applied societal importance of studying these skills. We present it as such on line 100.

* page 4, line 67: "Individuals typically need to succeed ..." - This sounds like it's referring to

university admission tests, which are not used everywhere in the world. I assume that the

authors are also referring more generally to any tests taken during secondary education that

influence people's chances of getting admitted to university, but this should be clarified.

Phrasing was adjusted to clarify that we are referring to general testing during secondary education and not American college entrance exams specifically (see lines 104 – 107). 

* page 5, line 115: I looked at the PIAAC sample items, which I had never seen before. What

struck me is that the sample items are very specific in that a clear question with a relatively

simple, unambiguous answer is asked before participants even start reading. The second item

is also not a text but a table. In both cases, 95% of the contents can be ignored by the reader if

they only want to answer the question. What I'm asking myself is how representative these

sample items are of PIAAC as a whole, and how similar the PIAAC items are to items used in,

say, a typical psycholinguistic experiment. This is relevant because the authors' contention is

that people who perform differently from the general population in PIAAC can't be reasonably

expected to perform similarly to the general population in a psycholinguistic experiment - but if

the two tasks mostly tap into unrelated skills, this conclusion would be less justified.

For more transparency, we have described a sample item on lines 230-231. In terms of how the test items relate to skills tested in psycholinguistic experiments, PIAAC tasks draw on multiple psycholinguistic processes at any given time. The assessment measures reading for a purpose (i.e., to achieve a goal such as gathering knowledge, forming an opinion, evaluating a text, etc.) while simulating the reading demands of everyday work and life. Reading for a purpose requires participants to draw on information processing skills and working memory capacity in addition to basic reading (decoding, vocabulary knowledge, word and sentence processing, inferential reasoning etc.) and numeric (addition, subtraction, quantitative reasoning etc.) skills (PIAAC Literacy Expert Group, 2009). PIAAC data do not provide direct access to the participant performance in these specific component skills (with an exception of a small subset of mostly low-literacy participants (OECD, 2021)). Since literacy and numeracy require combining and coordinating these subskills, performance on the PIAAC assessment can reasonably provide qualitative (if not quantitative) insight into how participants might perform on these component skills. We describe this reasoning in the introduction on lines 187 – 200. Additionally, in the discussion we clarify that this link is made cautiously and can only be used as an indicator that future research is warranted to determine the degree to which the undergraduate student bias affects these component skills directly (see lines 460 – 482). 

We also construe psycholinguistic experiments as a means to achieve understanding of the cognitive processes as they occur in the real-world tasks and environments, including reading for the purposes described above. One facet of such real-world tasks is navigating the wealth of information of which only a small fraction is relevant for the specific information-processing purpose. Of course, any single experiment, include the PIAAC tasks, only gives us a partial and limited glimpse of those processes, typically in a highly constrained environment. If it is the case, as the reviewer suggests, that strong populational differences observed in the more complex literacy and numeracy tasks may not appear in more constrained psycholinguistic experiments, this may signal a limitation in generalizability of those experiments to the real-world cognitive demands rather than validity of using the more complex/inclusive tasks. 

PIAAC Literacy Expert Group (2009). PIAAC Literacy: A Conceptual Framework. OECD Education Working Papers, No. 34, OECD Publishing. http://dx.doi.org/10.1787/220348414075

OECD (2021). The Assessment Frameworks for Cycle 2 of the Programme for the International Assessment of Adult Competencies. OECD Skills Studies. OECD Publishing: Paris. https://doi.org/10.1787/4bc2342d-en

* page 6: The "Statistical considerations" section needs some clarification. The section starts

with the introduction of "10 plausible values", which the reader doesn't know what to do with at

this point. There is some explanation later on, but it doesn't quite become clear to the

uninitiated reader what is happening and why the raw data can't be used.

Additional explanation has been added on lines 283 – 294. Raw data cannot be used because participants only respond to a subset of PIAAC test items. Plausible values were derived to estimate overall proficiency including on items that the participant did not respond to.

* page 8: Instead of Tables 1 and 2, I would have liked to see a histogram of the distribution of

literacy and numeracy scores in each sample. For instance, are the scores normally distributed?

This is crucial for interpreting mean and SD values. Furthermore, I don't quite see what showing

the scores for each country contributes to the main point.

We thank the reviewer for the suggestion. We have added Figs 1 and 2 which plot the distribution of literacy and numeracy skills respectively among the Student, Young and General samples of all countries combined to the main document (see lines 315-327). In the supporting information section we also include plots to show the distribution of skills for each country individually. Currently, we opted for leaving Tables 1 and 2 in the main paper, but we are prepared to move them to the supplementary materials if this is the Editor’s preference.

We feel it is important to include the breakdown of skills by country to allow other researchers to zoom in on the characteristics of these countries and how they may predict the given situation. While out of scope for this paper, it is likely that the socio-economic status of countries and their investment into post-secondary education co-determines effects sizes of the inter-sample differences under consideration. Reporting the by-country breakdown makes it possible for social scientists to pursue this question directly, without retreating to the raw PIAAC data, like we did. We make this future direction explicit on lines 488-492.

* page 10: I'm not sure that the comparison of effect sizes between PIAAC score differences

and psychology studies is warranted or useful. Even though Cohen's d is supposed to abstract

away from differences wrt *what* was measured on which scale, I think it's still apples and

oranges to compare differences in traits between groups of people with differences in

dependent variables caused by some experimental manipulation. I think this part could be

dropped without loss of clarity or relevance. By contrast, what I would like to see are the

estimates of the score differences between groups and their associated confidence intervals. In

order to interpret these, it would be helpful to know what the distribution of scores is (see

previous point).

Score differences themselves are not highly interpretable on the PIAAC assessment. For example, literacy and numeracy scores range from 0-500 points which are translated into 6 levels: below level 1 (0-175 points), level 1 (176-225 points), level 2 (226-275 points), level 3 (276-325 points), level 4 (326-375 points), and level 5 (376-500 points). Individuals scoring at level 2 are also expected to be proficient at level 1. It is not self-evident that reporting a difference in, say, 25 points is more informative than reporting a difference in (effectively) units of pooled standard deviation. Similarly, moving from standardized effect size estimates to score differences would lose the ability to compare the effect sizes observed in our data to those in other literature. Given the emphasis of the current statistical literature on reporting effect sizes (including statistical guidelines to this journal: https://journals.plos.org/plosone/s/submission-guidelines.#loc-statistical-reporting), we opted for retaining Cohen’s ds. Mathematically, Cohen’s d and confidence intervals of score differences are highly related in that they account for standard errors of the distributions under comparison, thus we do not feel that this decision masks any critical information from the readers.

We also respectfully disagree with the “oranges and apples” assessment of the application of Cohen’s d. Just like in the reviewer’s example, we report differences in dependent variables (literacy and numeracy scores), not in group or individual traits. Since seminal Cohen’s power primer (Cohen, 1992), the effect size has been proposed to apply to quantify differences between populations. This approach has also been used in numerous psycholinguistic experiments and meta-studies that compare different cohorts of participants, especially in bilingualism research (see a small selection of references below). Thus, we do not see an immediate reason for shying away from using an estimate of the effect size or from comparing it with the existing psycholinguistic literature. 

Schroeder, S. R. (2018). Do bilinguals have an advantage in theory of mind? A meta-analysis. Frontiers in Communication, 3, 36.

Malgady, R. G., & Costantino, G. (1998). Symptom severity in bilingual Hispanics as a function of clinician ethnicity and language of interview. Psychological Assessment, 10(2), 120.

Bird, E. K. R., Cleave, P., Trudeau, N., Thordardottir, E., Sutton, A., & Thorpe, A. (2005). The language abilities of bilingual children with Down syndrome.

Souza, A. L., Byers-Heinlein, K., & Poulin-Dubois, D. (2013). Bilingual and monolingual children prefer native-accented speakers. Frontiers in psychology, 4, 953.

* page 11: "We identified the magnitude of this bias ... " - This needs clarification. The authors

did not identify the magnitude of the undergraduate sampling bias, which to my mind would

indicate how much more likely undergraduates are to be sampled compared to other people. I

think what the authors mean is that they quantified the sampling bias regarding literacy and

numeracy skills that *results* from the undergraduate sampling bias.

We thank the reviewer for this point. We did not measure the magnitude of the bias, rather we quantified the degree to which undergraduate students represent other populations. Revisions made on line 416.

* References: Many references have incorrect capitalization, incomplete page numbers, and

other formatting issues.

Revisions have been made to address formatting issues and missing information. This section was also updated to reflect the revisions addressed in this letter.

Reviewer 2: 

This paper is clear and straightforward, raises an important point, and uses good

quality data and analyses. I have only minor queries. Frist, the description of the three different

groups on page 6 (line 141) makes it sound like participants could not be part of the 'young'

group and the 'general' group, but in actual fact I think from the description that these are

overlapping samples. This should be clarified. 

We thank the reviewer for a high evaluation of our paper. Participants could indeed be part of both the Young sample and the General sample, however both the Young and General sample are independent of the Student sample. Revisions made on lines 268 – 272 to clarify.

Second, I felt that is worth emphasising that these estimates have been calculated on the basis of single, relatively complex tasks. It is therefore not possible to know whether similar effect sizes exist for different types of literacy measure or numeracy measure. For example, it may be that group differences are reduced for simple tasks such as word reading, but the authors are not able to confirm this.

We have added a paragraph (lines 460 – 482) emphasising this limitation and while we predict that students will be unrepresentative in other measures, this question must ultimately be answered by future research.

---

## [Decision Letter · Decision Letter 1]

27 Jun 2022

How representative are student convenience samples? A study of literacy and numeracy skills in 32 countries

PONE-D-22-04101R1

Dear Dr. Wild,

We’re pleased to inform you that your manuscript has been judged scientifically suitable for publication and will be formally accepted for publication once it meets all outstanding technical requirements.

Kind regards,

Steven Frisson

Academic Editor

PLOS ONE

Additional Editor Comments (optional):

Thank you for addressing all the comments. I asked one of the original Reviewers to have another look at the manuscript and they (as well as I) are happy with the revision. I believe it's an important finding and I personally will not hesitate referring to it once it has been published!

Steven

Reviewers' comments:

Reviewer's Responses to Questions

**Comments to the Author**

1. If the authors have adequately addressed your comments raised in a previous round of review and you feel that this manuscript is now acceptable for publication, you may indicate that here to bypass the “Comments to the Author” section, enter your conflict of interest statement in the “Confidential to Editor” section, and submit your "Accept" recommendation.

Reviewer #1: All comments have been addressed

2. Is the manuscript technically sound, and do the data support the conclusions?

Reviewer #1: Yes

3. Has the statistical analysis been performed appropriately and rigorously? 

Reviewer #1: Yes

4. Have the authors made all data underlying the findings in their manuscript fully available?

Reviewer #1: Yes

5. Is the manuscript presented in an intelligible fashion and written in standard English?

Reviewer #1: Yes

6. Review Comments to the Author

Reviewer #1: The revision has addressed all my comments on the previous version. I can therefore recommend publication.

7. PLOS authors have the option to publish the peer review history of their article (what does this mean?). If published, this will include your full peer review and any attached files.

Reviewer #1: No

---

## [Editor Report · Acceptance letter]

29 Jun 2022

PONE-D-22-04101R1 

How representative are student convenience samples? A study of literacy and numeracy skills in 32 countries 

Dear Dr. Wild:

I'm pleased to inform you that your manuscript has been deemed suitable for publication in PLOS ONE. Congratulations! Your manuscript is now with our production department. 

Kind regards, 

on behalf of

Dr. Steven Frisson 

Academic Editor

PLOS ONE